# ALGONET: $C^\infty$ SMOOTH ALGORITHMIC NEURAL NETWORKS

## ABSTRACT

Artificial neural networks have revolutionized many areas of computer science in recent years, providing solutions to a number of previously unsolved problems. On the other hand, for many problems, classic algorithms exist, which typically exceed the accuracy and stability of neural networks. To combine these two concepts, we present a new kind of neural networks—algorithmic neural networks (AlgoNets). These networks integrate smooth versions of classic algorithms into the topology of neural networks. A forward AlgoNet includes algorithmic layers into existing architectures to enhance performance and explainability while a backward AlgoNet enables solving inverse problems without or with only weak supervision. In addition, we present the `algonet` package, a PyTorch based library that includes, inter alia, a smoothly evaluated programming language, a smooth 3D mesh renderer, and smooth sorting algorithms.

## 1 INTRODUCTION

Artificial Neural Networks are employed to solve numerous problems, not only in computer science but also in all other natural sciences. Yet, the reasoning for the topologies of neural networks seldom reaches beyond empirically-based decisions.

In this work, we present a novel approach to designing neural networks—algorithmic neural networks (short: AlgoNet). Such networks integrate algorithms as algorithmic layers into the topology of neural networks. However, propagating gradients through such algorithms is problematic, because crisp decisions (conditions, maximum, etc.) introduce discontinuities into the loss function. If one passes from one side of a crisp decision to the other, the loss function may change in a non-smooth fashion—it may "jump." That is, the loss function suddenly improves (or worsens, depending on the direction) without these changes being locally noticeable anywhere but exactly at these "jumps." Hence, a gradient descent based training, regardless of the concrete optimizer, cannot approach these "jumps" in a systematic fashion, since neither the loss function nor the gradient provides any information about these "jumps" in any place other than exactly the location at which they occur. Therefore, a smoothing is necessary, such that information about the direction of improvement becomes exploitable by gradient descent also in the area surrounding the "jump." That is, by smoothing, e.g., an `if`, one can smoothly, by gradient descent, undergo a transition between the two crisp cases using only local gradient information.

Generally, for end-to-end trainable neural network systems, all components should at least be $C^0$ smooth, i.e., continuous, to avoid "jumps." However, having $C^k$ smooth, i.e., $k$ times differentiable and then still continuous components with $k \geq 1$ is favorable. This property of higher smoothness allows for higher-order derivatives and thus prevents unexpected behavior of the gradients. Hence, we designed smooth approximations to basic algorithms where the functions representing the algorithms are ideally $C^\infty$ smooth. That is, we designed pre-programmed neural networks (restricted to smooth components) with the structure of given algorithms.

Algorithmic layers can solve sub-problems of the given problem, act as a custom algorithmic loss, or assist in finding an appropriate solution for (ill-posed) inverse problems. Such algorithmic losses can impose constraints on predicted solutions through optimization with respect to the algorithmic loss. Ill-posed problems are a natural application for algorithmic losses and algorithmic layers. For that, we introduce the Reconstructive Adversarial Network (RAN) in Sec. 3.3.

In this work, we describe the basic concept of algorithmic layers and present several applications. In Sec. 3.1.1, we start by proving that any algorithm, which can be emulated by a Turing machine, can be approximated by a $C^\infty$ smooth function. In Sec. 4, we present some algorithmic layers that we designed to solve underlying problems. In the appendix, we present a case study on 3D geometry reconstruction that demonstrates the applicability of RANs. All described algorithmic layers and models are provided in the `algonet` package, a PyTorch (Paszke et al. (2017)) based library for AlgoNets.

## 2 RELATED WORK

Related work (Mart'in Abadi et al. (2015); Che et al. (2018); Henderson & Ferrari (2018)) in neural networks focused on dealing with crisp decisions by passing through gradients for the alternatives of the decisions. There is no smooth transition between the alternatives, which introduces discontinuities in the loss function that hinder learning, which of the alternatives should be chosen. TensorFlow contains a sorting layer (`tf.sort`) as well as a while loop construct (`tf.while_loop`). Since the sorting layer only performs a crisp relocation of the gradients and the while loop has a crisp exit condition, there is no gradient with respect to the conditions in these layers. Concurrently, we present a smooth sorting layer in Sec. 4.1 and a smooth while loop in Sec. 3.1.1.

Theoretical work by DeMillo *et al.* (DeMillo & Lipton (1993)) proved that any program could be modeled by a smooth function. Consecutive works (Nesterov (2005); Chaudhuri & Solar-Lezama (2011); Yang & Barnes (2017)) provided approaches for smoothing programs using, i.a., Gaussian smoothing (Chaudhuri & Solar-Lezama (2011); Yang & Barnes (2017).)

## 3 ALGONET

To introduce algorithmic layers, we prove that smooth approximations for any Turing computable algorithm exist and explain two flavors of AlgoNets: forward and backward AlgoNets.

### 3.1 SMOOTH ALGORITHMS

To design a smooth algorithm, all discrete cases (e.g., conditions of `if` statements or loops) have to be replaced by continuous or smooth functions. The essential property is that the implementation is differentiable with respect to all internal choices and does not—as in previous work—only carry the gradients through the algorithm. For example, an `if` statement can be replaced by a sigmoid-weighted sum of both cases. By using a smooth sigmoid function, the statement is smoothly interpreted. Hence, the gradient descent method can influence the condition to hold if the content of the `then` case reduces the loss and influence the condition to fail if the loss is lower when the `else` case is executed. Thus, the partial derivative with respect to a neuron is computed because the neuron is used in the `if` statement. In contrast, when propagating back the gradient of the `then` or the `else` case depending on the value of the condition, there is a discontinuity at the points where the value of the condition changes and the partial derivative of the neuron in the condition equals zero.

$$s_1(x, s) = \frac{1}{1 + e^{-x \cdot s}} \quad \text{with } s = 1 \quad (1) \qquad s_2(x) = \begin{cases} 0 & \text{if } x < 0 \\ 1 & \text{else} \end{cases} \quad (2)$$

Here, the logistic sigmoid function (Eq. 1) is a $C^\infty$ smooth replacement for the Heaviside sigmoid function (Eq. 2), which is equivalent to the `if` statement. Alternatively, one could use other sigmoid functions, e.g., the $C^1$ smooth step function $x^2 - 2 \cdot x^3$ for $x \in [0, 1]$, and 0 and 1 for all values before and after the given range, respectively.

Another example is the max-operator, which, in neural networks, is commonly replaced by the SoftMax operator $\left( \text{SoftMax}(\mathbf{x})_i = \frac{\exp(x_i)}{\sum_j \exp(x_j)} \right)$.

After designing an algorithmic layer, we can use it to enhance a neural network and to solve for its inverse by using the reconstructive adversarial neural network (RAN) as shall be described in Sec. 3.2, and 3.3.

### 3.1.1 SMOOTH WHILE-PROGRAMS

In this section, we prove that for any algorithm, $C^\infty$ smooth approximations exist. For that, we present smooth and differentiable approximations to an elementary programming language based on the WHILE-language by Uwe Schöning (Schöning (2008).) We do not want to imply that such a direct translation of any algorithm into the WHILE-language to make it smooth is the best or even the only option. It is rather meant as a fallback solution if no better solution can be found for a sub-problem. We will present several more efficient and canonical approximations in Sec. 4.

The WHILE-language is Turing-complete, and for variables (`var`) its grammar can be defined as follows:

```
prog = WHILE var != 0 DO prog END
     | prog prog
     | var := var                  //  left and right var unequal
     | var := var + 1              //  left and right var equal
     | var := var - 1              //  left and right var equal
```

Here, `var` $\in \{$`xn` $\mid$ n $\in \mathbb{N}_0\}$ and while `x0` is the output, `xn` where n $\in \mathbb{N}_0$ are variables initialized to 0 if not set to an input value. Thus, $\{$`xn` $\mid$ n $\in \mathbb{N}_+\}$ are inputs and/or local variables used in the computations. Although this interpretation allows only for a single scalar output value (`x0`), an arbitrary number of output values can be reached, either by interpreting additional variables as output or by using multiple WHILE-programs (one for each output value). Provided Church's thesis holds, this language covers all effectively calculable functions. The WHILE-language is equivalent to register machines which have no WHILE, but instead IF and GOTO statements.

We generate the approximation to this language by executing all statements only to the extent of their probability. That is, we keep track of a probability $p$, indicating whether the current statement is still executed. Initially, $p = 1$. In the body of a while loop, the probability for an execution is $p_k^{(\text{new})}(x) = p_k^{(\text{old})} \cdot \phi_k(x)$ where $k$ defines the smoothness of the probability function for exiting the loop $\phi_k$. To obtain $C^\infty$ smooth

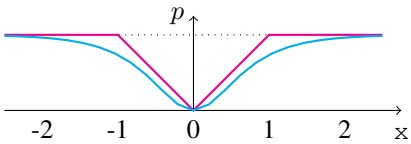

Figure 1: $\phi_0$ (magenta) and $\phi_\infty$ (cyan).

WHILE-programs, we used $\phi_\infty : \mathbb{R} \to [0,1] : x \mapsto \frac{(e^{sx}-1)^2}{e^{2sx}+1} = 1 - \text{sech}(sx)$. For $C^0$ smoothness, we used the shouldered fuzzy set $\phi_0 : \mathbb{R} \to [0,1] : x \mapsto 1 - \max(0, 1 - |x|) = \min(1, |x|)$. For `x0`...`xn` initialized as integers, using $\phi_0$, the result for the $C^0$ WHILE-program always equals the result for the discrete WHILE-program since the probability is always either 1 or 0. For all $k \in \mathbb{N}_0$, $\phi_k$ exists. Fig. 1 shows how these exit probability functions behave. Because of the symmetry of $\phi$, w.l.o.g., we can assume that $\forall x \in \mathbb{R}_{\geq 0}$. If the loop (in the discrete version) increases `x` by 1, `x` will diverge. If `x` decreases by 1, $p$ converges to 0. Since $\phi_0(x) \geq \phi_\infty(x)$, it suffices to show that $p$ converges to zero for $\phi_0$. Since $p^{(\text{new})}(x) = p^{(\text{old})}(x) \cdot \phi_0(x) \leq \phi_0(x) \leq |x|$, `x` := `x` $- p^{(\text{new})}(x) \geq$ `x` $- |x| = 0$. Thus, `x` and $\phi_0(x)$ monotonically decrease and `x` $\geq 0$. Eventually, $\phi_0(x) < 1$. Thus, $p(x) \leq (\phi_0(x))^n \xrightarrow{n \to \infty} 0$.

Here, $x$ is the value of the current variable, $s$ is the steepness, and $p$ is the probability of the execution. To apply the probabilities on the assignment, increment and decrement operators, we redefine them as:

$$\text{x0} := \text{x1} \quad \to \quad \text{x0} := p \cdot \text{x1} + (1-p) \cdot \text{x0} \tag{3}$$
$$\text{x0} := \text{x0} + 1 \quad \to \quad \text{x0} := \text{x0} + p \tag{4}$$
$$\text{x0} := \text{x0} - 1 \quad \to \quad \text{x0} := \text{x0} - p \tag{5}$$

Since the hyperbolic secant is $C^\infty$ smooth, our version of the WHILE-language is $C^\infty$ smooth. While the probability converges to zero, it (in most cases) never reaches zero, and the loop would never exit. Thus, we introduce $\epsilon > 0$ and exit the loop if $p \leq \epsilon$ or a maximum number of iterations is reached. Although this introduces discontinuities, by choosing an $\epsilon$ of numerically negligible size, the discontinuities also become numerically negligible.

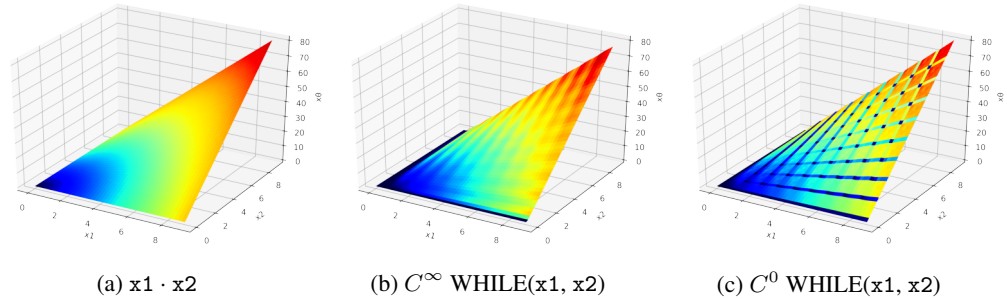

(a) x1 · x2          (b) $C^\infty$ WHILE(x1, x2)          (c) $C^0$ WHILE(x1, x2)

Figure 2: Multiplication of x1 and x2 implemented as a general multiplication, $C^\infty$ smooth WHILE-program and $C^0$ smooth WHILE-program. The height indicates the result of the function. Contrary to the common notion, the color indicates not the values but the analytic gradient of the function.

As an experiment, we implemented the multiplication on positive integers, as shown on the left:

```
WHILE x2 != 0 DO          | WITH p₁ := 1;  p₁' := p₁ · φ(x2) DO
    x3 := x1              |     x3  := p₁ · x1 + (1 − p₁) · x3
    WHILE x3 != 0 DO      |     WITH p₂ := p₁;  p₂' := p₂ · φ(x3) DO
        x0 := x0 + 1      |         x0  := x0 + p₂
        x3 := x3 − 1      |         x3  := x3 − p₂
    END                  |     WHILE p₂ ≥ ε
    x2 := x2 − 1         |     x2  := x2 − p₁
END                      | WHILE p₁ ≥ ε
```

Contrary to the discrete implementation, the smooth interpretation (as on the right) can interpolate the result for arbitrary values $x1, x2 \in \mathbb{R}_+$.

Since the WHILE-language is Turing-complete, any high-level program could, in principle, using an appropriate compiler, be translated into an equivalent program in WHILE-language. To this WHILE-program, automatic smoothing could be applied using the rules that are illustrated here. Of course, there are better ways of smoothing: manual smoothing using domain-specific knowledge and smoothing using a higher-level language outperform the low-level automatic smoothing. For example, in a higher-level language, the multiplication would be implemented since it is smooth itself. Using domain-specific knowledge, algorithms could be reformulated in such a way that smoothing is possible in a more canonical way. To translate a WHILE-program into a neural network, the WHILE loops are considered as recurrent sub-networks.

### 3.2 FORWARD ALGONET

The AlgoNet can be classified into two flavors, the forward and the backward AlgoNet. To create a forward AlgoNet, we use algorithmic layers and insert them into a neural network. By doing so, the neural network may or may not find a better local minimum by additionally employing the given algorithm. We do so by using one of the following options for each algorithmic layer:

- Insert between two consecutive layers (Fig. 3a).
- Insert between two consecutive layers and also skip the algorithmic layer (Fig. 3b).
- Add a residual connection and apply the algorithmic layer on the residual part (Fig. 3c).

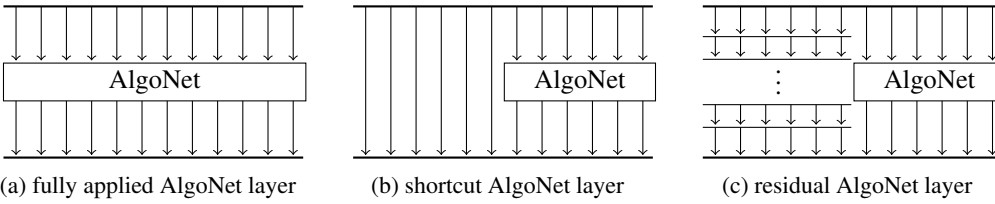

(a) fully applied AlgoNet layer          (b) shortcut AlgoNet layer          (c) residual AlgoNet layer

Figure 3: Different styles of the forward AlgoNet.

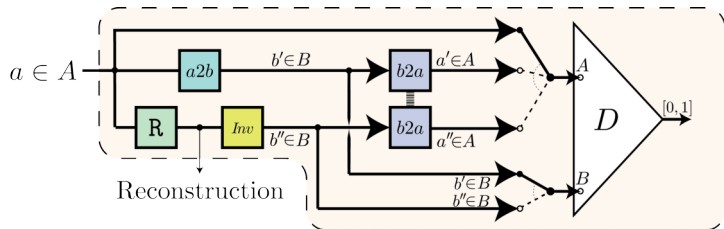

Figure 4: RAN System overview. The reconstructor receives an object from the input domain $A$ and predicts the corresponding reconstruction. The reconstruction, then, is validated through our smooth inverse. The latter produces objects in a different domain, $B$, which are translated back to the input domain $A$ for training purposes ($b2a$). Unlike in traditional GAN systems, the purpose of our discriminator $D$ is mainly to indicate whether the two inputs match in content, not in style. Our novel training scheme trains the whole network via five different data paths, including two which require another domain translator, $a2b$.

Generally, algorithmic layers do not have trainable weights. Regarding the accuracy, the output of $C^\infty$ smooth WHILE-programs differs from the discrete WHILE-programs by a small factor and offset. The output of $C^0$ smooth WHILE-programs equals the output of discrete WHILE-programs for integer inputs (discrete WHILE-programs fail for non-integer inputs). One could counter this factor and offset for $C^\infty$ smooth WHILE-programs by adding an additional weight and a bias to each assignment in the WHILE-program. For that, one should regularize these weights and biases to be close to one and zero, respectively. These parameters could be trained on a data set of integer input/output pairs of the respective discrete WHILE-program to fit the algorithmic layer. Moreover, the algorithmic layer could also be trained to fit the surrounding layers better.

### 3.3 BACKWARD ALGONET: RECONSTRUCTIVE ADVERSARIAL NETWORKS (RAN)

While forward AlgoNets can use arbitrary smooth algorithms—of course, an algorithm directly related to the problem might perform better—backward AlgoNets use an algorithm that solves the inverse of the given problem. For example, a smooth renderer for 3D-reconstruction, a smooth iterated function system (IFS) for solving the inverse-problem of IFS, and a smooth text-to-speech synthesizer for speech recognition. While backward AlgoNets could be used in supervised settings, they are designed for unsupervised or weakly supervised solving of inverse-problems. Their concept is the following:

Input ($\in A$) $\to$ `Reconstructor` $\to$ Goal $\to$ `smooth inverse` $\to$ Smooth version of input ($\in B$)

This structure is similar to auto-encoders and the encoder-renderer architecture presented by Che *et al.* (Che et al. (2018).) Such an architecture, however, cannot directly be trained since there is a domain shift between the input domain $A$ and the smooth output domain $B$. Thus, we introduce domain translators ($a2b$ and $b2a$) to translate between these two domains. Since training is extremely hard with three consecutive components, of which the middle one is highly restrictive, we introduce a novel training schema for these components: the reconstructive adversarial network (RAN). For that, we also include a discriminator to allow for adversarial training of the components $a2b$ and $b2a$. Of our five components four are trainable (the reconstructor R, the domain translators $a2b$ and $b2a$, and the discriminator $D$), and one is non-trainable (the smooth inverse *Inv*).

Since, initially, neither the reconstructor nor the domain translators are trained, we are confronted with a causality dilemma. A typical approach for solving such causality dilemmas is to solve the two components coevolutionarily by iteratively applying various influences towards a common solution. Fig. 4 depicts the structure of the RAN, which allows for such a coevolutionary training scheme.

The discriminator receives two inputs, one from space $A$ and one from space $B$. One of these inputs (either $A$ or $B$) receives two values, a real and a fake value; the task of the discriminator is to distinguish between these two, given the other input. For training, the discriminator is trained to distinguish between the different path combinations for the generation of inputs. Consequently, the generator modules are trained to fool the discriminator. This adversarial game allows training the RAN.

In the following, we will present this process, as well as its involved losses, in detail. Our optimization of $R$, $a2b$, $b2a$, and $D$ involves adversarial losses, cycle-consistency losses, and regularization losses. Specifically, we solve the following optimization:

$$\min_{R} \min_{a2b} \min_{b2a} \max_{D} \mathcal{L} \quad \text{or in greater detail} \quad \min_{R} \min_{a2b} \min_{b2a} \max_{D} \sum_{i=1}^{5} (\alpha_i \cdot \mathcal{L}_i) + \mathcal{L}_{\text{reg}}.$$

where $\alpha_i$ is a weight in $[0, 1]$ and $\mathcal{L}$, and $\mathcal{L}_i$ shall be defined below. $\mathcal{L}_{\text{reg}}$ denotes the (optional) regularization losses imposed on the reconstruction output.

We define $b', b'' \in B$ and $a', a'' \in A$ in dependency of $a \in A$ according to Fig. 4 as

$$b' = a2b(a) \qquad b'' = \textit{Inv} \circ R(a) \qquad a' = b2a(b') \qquad a'' = b2a(b'').$$

With that, our losses are (without hyper-parameter weights)

$$\mathcal{L}_1 = \mathbb{E}_{a \sim A}[\log D(a, b'')] + \mathbb{E}_{a \sim A}[\log(1 - D(a, b'))] + \mathbb{E}_{a \sim A}[\|b'' - b'\|_1]$$

$$\mathcal{L}_2 = \mathbb{E}_{a \sim A}[\log D(a, b'')] + \mathbb{E}_{a \sim A}[\log(1 - D(a'', b''))] + \mathbb{E}_{a \sim A}[\|a'' - a\|_1]$$

$$\mathcal{L}_3 = \mathbb{E}_{a \sim A}[\log D(a, b')] + \mathbb{E}_{a \sim A}[\log(1 - D(a'', b'))] + \mathbb{E}_{a \sim A}[\|a' - a\|_1] + \mathbb{E}_{a \sim A}[\|b'' - b'\|_1]$$

$$\mathcal{L}_4 = \mathbb{E}_{a \sim A}[\log D(a, b'')] + \mathbb{E}_{a \sim A}[\log(1 - D(a', b''))] + \mathbb{E}_{a \sim A}[\|a' - a\|_1] + \mathbb{E}_{a \sim A}[\|b'' - b'\|_1]$$

$$\mathcal{L}_5 = \mathbb{E}_{a \sim A}[\log D(a, b')] + \mathbb{E}_{a \sim A}[\log(1 - D(a', b'))] + \mathbb{E}_{a \sim A}[\|a' - a\|_1].$$

We alternately train the different sections of our network in the following order:

1. The discriminator $D$
2. The translation from $B$ to $A$ ($b2a$)
3. The components that perform a translation from $A$ to $B$ ($R$+$\textit{Inv}$, $a2b$)

For each of these sections, we separately train the five losses $\mathcal{L}_1, \mathcal{L}_2, \mathcal{L}_3, \mathcal{L}_4$, and $\mathcal{L}_5$. In our experiments, we used one Adam optimizer (Kingma & Ba (2014)) for each trainable component ($R$, $a2b$, $b2a$, and $D$).

## 4 APPLICATIONS

In this section, we present specific AlgoNet-layers. Specifically, we present a smooth sorting algorithm, a smooth median, a finite differences layer, a weighted SoftMax, smooth iterated function systems, and a smooth 3D mesh renderer. We will present the smooth 3D mesh renderer in greater detail in the appendix, where we will also present the respective 3D mesh reconstruction applying the RAN.

### 4.1 SOFTSORT

The SoftSort layer is a smooth sorting algorithm that is based on a parallelized version of bubble sort (Knuth (1998)) (see especially Section 5.3.4: Networks for Sorting), which sorts a tensor along an array of scalars by repeatedly exchanging adjacent elements if necessary. Fig. 5 shows the structure of the SoftSort algorithm. Contrasting our approach, the sorting layer in TensorFlow (Mart'in Abadi et al. (2015)) is not smooth and does not consider gradients with respect to the ordering induced by the sorting.

### 4.2 FINITE DIFFERENCES

The finite differences method, which was introduced by Lewy *et al.* (Lewy H. (1928)), is an essential tool for finding numerical solutions of partial differential equations. In analogy, the finite differences layer uses finite differences to compute the spatial derivative for one or multiple given dimensions of a tensor. For that, we subtract the tensor from itself shifted by one in the given dimension. Optionally, we normalize the result by shifting the mean to zero and/or add padding to output an equally sized tensor. Thus, it is possible to integrate a spatially- or temporally-derivating layer into neural networks.

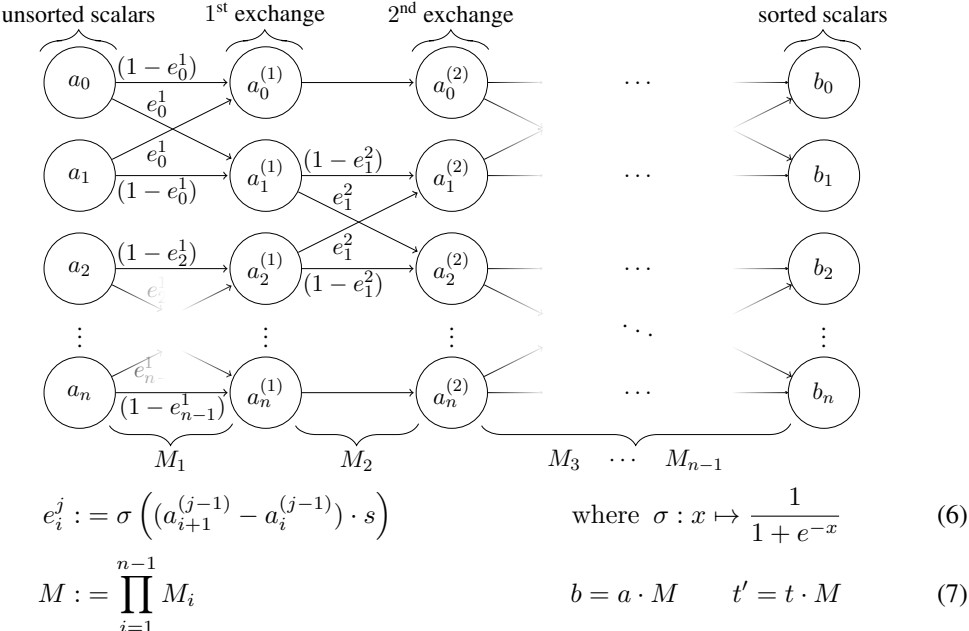

$$e_i^j := \sigma\left((a_{i+1}^{(j-1)} - a_i^{(j-1)}) \cdot s\right) \qquad \text{where } \sigma : x \mapsto \frac{1}{1 + e^{-x}} \qquad (6)$$

$$M := \prod_{i=1}^{n-1} M_i \qquad\qquad b = a \cdot M \qquad t' = t \cdot M \qquad (7)$$

Figure 5: The structure of SoftSort. Here, the exchanges of adjacent elements are represented by matrices $M_i$. By multiplying these matrices with tensor $a$, we obtain $b$, the sorted version of $a$. By instead multiplying with tensor $t$, we obtain $t'$: $t$ sorted with respect to $a$. Using that, we can also sort a tensor with respect to a learned metric. For sorting $n$ values, we need $n - 1$ steps for an even $n$ and $n$ steps for an odd $n$; to get a probabilistic coarse sorting, even fewer steps may suffice. $s$ denotes the steepness of the sorting such that for $s \to \infty$ we obtain a non-smooth sorting and for infinitely many sorting operations, all resulting values equal the mean of the input tensor. In the displayed graph, the two recurrent layers are unrolled in time.

### 4.3 WEIGHTEDSOFTMAX

The weighted SoftMax (short: $\mathfrak{w}$SoftMax) allows a list that is fed to the SoftMax operator to be smoothly sliced by weights indicating which elements are in the list. That is, there are two inputs, the actual values ($\mathbf{x}$) and weights ($w$) from $(0; 1]$ indicating which values of $\mathbf{x}$ should be considered for the SoftMax. Thus, $\mathfrak{w}$SoftMax can be used when the maximum value of values, for which an additional condition also holds, is searched by indicating whether the additional condition holds with weights $w_i \in (0; 1]$. We define the weighted SoftMax as:

$$\mathfrak{w}\text{SoftMax}_i(\mathbf{x}, w) := \frac{\exp(\mathbf{x}_i) \cdot w_i}{\sum_{i=0}^{\|w\|-1} \exp(\mathbf{x}_i) \cdot w_i} = \frac{\exp(\mathbf{x}_i + \log w_i)}{\sum_{i=0}^{\|w\|-1} \exp(\mathbf{x}_i + \log w_i)} \qquad (8)$$
$$= \text{SoftMax}_i(\mathbf{x}_i + \log w_i)$$

Accordingly, we define the weighted SoftMin (analogue to SoftMax/SoftMin) as $\mathfrak{w}$SoftMin$(\mathbf{x}, w) := \mathfrak{w}$SoftMax$(-\mathbf{x}, w)$. By that, we enable a smooth selection to apply the SoftMax/SoftMin function only to relevant values.

### 4.4 SOFTMEDIAN

The mean is a commonly used measure for reducing tensors, e.g., for normalizing a tensor. While the median is robust against outliers, the mean is sensitive to all data points. This has two effects: firstly, the mean is not the most representative value because it is influenced by outliers; secondly, the derivative of a normalization substantially depends on the positions of outliers. That is, outliers, which might have accelerated gradients in the first place, can influence all values during a normalization like $x' := x - \bar{x}$. While one would generally avoid these potentially malicious gradients by cutting the gradients of $\bar{x}$, this is not adequate if changes in $\bar{x}$ are expected. To reduce the influence of outliers in a smooth way, we propose the SoftMedian, which comes in two styles, a precise and slower as well as a significantly faster version that only discards a fixed number of outliers.

The precise version sorts the tensor with SoftSort and takes the middle value(s). For that, it is not necessary to carry out the entire SoftSort; instead, only those computations that influence the middle value need to be taken into account.

The faster variant to compute the SoftMedian (of degree $j$) is by its recursive definition in which influence of the minimum and maximum values is reduced as follows: $\text{SoftMed}(\mathbf{x})^{(j)} :=$ $\mathfrak{w}\text{SoftMin}\left(\mathfrak{w}\text{SoftMin}(\mathbf{x}, \text{SoftMed}(\mathbf{x})^{(j-1)}) + \mathfrak{w}\text{SoftMax}(\mathbf{x}, \text{SoftMed}(\mathbf{x})^{(j-1)}), \text{SoftMed}(\mathbf{x})^{(j-1)}\right)$ where $\text{SoftMed}(\mathbf{x})^{(0)} := \|\mathbf{x}\|$.

### 4.5 Smooth Iterated Function Systems

Iterated function systems (IFS) allow the construction of various fractals using only a set of parameters. For example, pictures of plants like Barnsley's fern can be generated using only $4 \times 6 = 24$ parameters. Numerous different plants and objects can be represented using IFS. Since IFS are parametric representations, they can be stored in very small space and be adjusted. This can be used, e.g., in a computer game to avoid unnatural uniformity when rendering vegetation by changing the parameters slightly, so that each plant looks slightly different. Finally, there are very fast algorithms to generate images from IFS. While IFS provide many advantages, solving the inverse problem of IFS, i.e., finding an IFS representation for any given image, is very hard and still unsolved. Towards solving this inverse problem, we developed a $C^\infty$ smooth approximation to IFS.

Given a two-dimensional IFS with $n$ bi-linear functions $(f_i)_{i \in \{1..n\}}$ $f_i(x,y) := (x + a_{1,i} + a_{2,i}x + a_{3,i}y, y + a_{4,i} + a_{5,i}x + a_{6,i}y)$, we repeatedly randomly select one function $f \in (f_i)_{i \in \{1..n\}}$ and apply it to an initial position or the proceeding result. We do that process arbitrarily often and plot every intermediate step. Since it is not meaningful to interpolate multiple functions, because IFS rely on randomized choices, and to provide consistency, we perform these random choices in advance.

The difficulty in this process is the rasterization since no crisp decision correlating pixels to points can be made. Thus, we correlate each pixel to each point with a probability $p \in [0; 1]$ where $p = 1$ is a full correlation and $p = 0$ means no correlation at all. By applying Gaussian smoothing on the locations of the points, for each point, the probabilities $p \in (0; 1)$ for each pixel define the correlation. Concluding, for each pixel, the probabilities for all points to lie in the area of that pixel are known. By aggregating these probabilities, we achieve a smooth rasterization.

We tested the smooth IFS by optimizing its parameters to fit an image. For that, by setting the standard deviation, different levels of details can be optimized.

### 4.6 Smooth Renderer

Lastly, we include a $C^\infty$ smooth 3D mesh renderer to the AlgoNet library, which projects a triangular mesh onto an image while considering physical properties like perspective and shading. Compared to previous differentiable renderers, this renderer is fully and not only locally differentiable. Moreover, the continuity of the gradient allows for seamless integration into neural networks by avoiding unexpected behavior altogether. By taking the decision which triangles cover a pixel, in analogy to the smooth rasterization in Sec. 4.5, the silhouette of the mesh can be obtained. Consecutively, by computing which of these triangles is the closest to the camera smoothly, our renderer's depth buffer is smooth. That allows for color handling and shading.

The smooth renderer is presented in greater detail in Appendix A. Consecutively, in Appendix B, we use the smooth renderer as smooth inverse for the RAN to solve the inverse problem of 3D reconstruction. A selection of results is presented in Fig. 6; more results are presented in Appendix C.

## 5 Discussion and Conclusion

We presented AlgoNets as a new kind of layers for neural networks, a $C^\infty$ Turing complete interpreter, and RANs as a novel technique for solving ill-posed inverse problems. Moreover, in the appendix, we demonstrate with a case study on 3D reconstruction that the RAN works even in complex settings. We have implemented the presented layers on top of PyTorch and will publish our AlgoNet library upon publication of this work. Concurrent with their benefits, AlgoNets, such as the aforementioned rendering layer, can get computationally very expensive. On the other hand, the rendering layer is

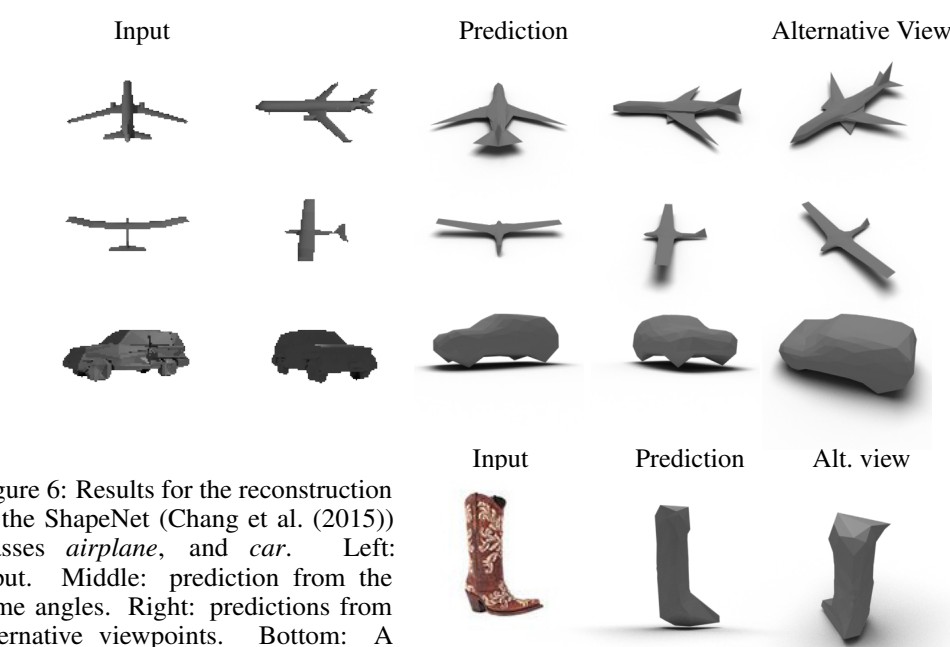

Figure 6: Results for the reconstruction of the ShapeNet (Chang et al. (2015)) classes *airplane*, and *car*. Left: input. Middle: prediction from the same angles. Right: predictions from alternative viewpoints. Bottom: A single-view reconstruction result from the UT Zappos50K dataset (Yu & Grauman (2014)) (camera-captured images). Although the object has strong textures, it is adequately reconstructed.

very powerful since it allows training a 3D reconstruction without 3D supervision using the RAN. Since the RAN has a very complex architecture that requires a particular training paradigm, it can also take relatively long to train it. To accommodate this issue, we found that by increasing some loss weights and introducing a probability of whether a computation is executed, the training time can be reduced by a factor of two or more. Unfortunately, due to length restrictions, we have only been able to present a subset of all implemented algorithmic layers.

Rectified Linear Units (ReLUs) (Nair & Hinton (2010)) have been very successful in practice for efficient back-propagation. However, in contrast to such a network where the rectification allows to activate or deactivate neurons, we use the gradients produced by the differentiable algorithm not just for adapting parameters but employ the gradient of the smooth algorithm to train other parts of the network. For doing so, the gradient of the algorithm should not be discontinuous (as it would be with a ReLU); so having at least a $C^1$ smoothness is clearly beneficial to allow for effective training.

The AlgoNet could also be used in the realm of explainable artificial intelligence (Gilpin et al. (2018)) by adding residual algorithmic layers into neural networks and then analyzing the neurons of the trained AlgoNet. For that, network activation and/or network sensitivity can indicate the relevance of the residual algorithmic layer. To compute the network sensitivity of an algorithmic layer, the gradient with respect to additional weights (constant equal to one) in the algorithmic layer could be computed. By that, similarities between classic algorithms and the behavior of neural networks could be inferred. An alternative approach would be to gradually replace parts of trained neural networks with algorithmic layers and analyzing the effect on the new model accuracy.

In the future, we will develop a high-level smooth programming language to improve smooth representations of higher-level programming concepts. Adding trainable weights to the algorithmic layers to improve the accuracy of smooth algorithms and/or allow the rest of the network to influence the behavior of the algorithmic layer is subject to future research. The similarities of our smooth WHILE-programs to analog computing as well as quantum computing shall be explored in future work. Another future objective is the exploration of neural networks not with a fixed but instead a smooth topology. Finally, loss ranking would be an additional application for the SoftSort algorithm.

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

## ADDITIONAL MATERIAL: A CASE STUDY ABOUT 3D RECONSTRUCTION

Here, we describe how we use the AlgoNet to solve a complex problem in computer graphics without supervision: the reconstruction of 3D geometry from single real-world images. This problem requires a complex system that can only be implemented by including algorithmic concepts provided by the AlgoNet and the RAN.

In Appendix A, we present our $C^\infty$ smooth differentiable 3D mesh renderer in greater detail. Based on that, we present unsupervised 3D geometry reconstruction as an application of the RAN in Appendix B. We present and discuss our 3D geometry reconstruction results in Appendix C. Finally, in Appendix D, we present additional implementation details.

## A $C^\infty$ SMOOTH RENDERER

In this section, we present our $C^\infty$ Smooth Renderer that avoids any discontinuities at occlusions or dis-occlusions. Having this property, the renderer's back-propagated gradients can be properly used to modify the 3D model. This is critical for integrating the renderer into a neural network. The typical discontinuity problem occurs during triangle rasterization, where the visibility of a triangle, due to occlusion or dis-occlusion, causes an abrupt change in the image. For example, if during the optimization process, the backside of a predicted object self-intersects its front, traditional differentiable renderers are not able to provide a reasonable gradient towards reversing such an erroneous self-intersection since they cannot differentiate with respect to occlusion. To overcome this problem, our approach offers a soft blending scheme, that is continuous even through such intersections.

As in the general rendering approach, first, we apply view transformations on all triangles to bring them from object space into perspective projection space coordinates. This process is generally already fully differentiable.

Consecutively, one needs rasterization to correlate triangles to pixels. General rasterization consists of two steps, for each pixel one needs to collect all the triangles that cover that pixel, and then employ a z-buffer to determine which of them is visible in the pixel.

Instead of collecting all triangles that fit the xy-coordinates of a given pixel, we determine a probability value of whether a triangle fits a pixel for each triangle and pixel. This constitutes the visibility tensor $V$ as shall be described in section A.2.

Our key idea is to use a visibility test that enables reasoning beyond occlusion, using only smooth functions to avoid abrupt changes. Rather than taking a discrete decision of which triangle is the closest and thus visible, we softly blend their visibility, which goes along with an idea from stabilizing non-photorealistic rendering results (Luft et al. (2006)). By using a SoftMin-based function, we determine the closest and thus most visible face. But using the simple SoftMin of the z-positions in camera space would result in only the single closest triangle being most visible. Thus, we need to incorporate the visibility tensor $V$ that tells us which triangles cover a given pixel. Instead, we weight the SoftMin with the visibility tensor $V$ by introducing the weighted SoftMin (ɯSoftMin). Taking the ɯSoftMin of the z-positions in camera space, constitutes smooth z-buffer as shall be described in greater detail in section A.1.

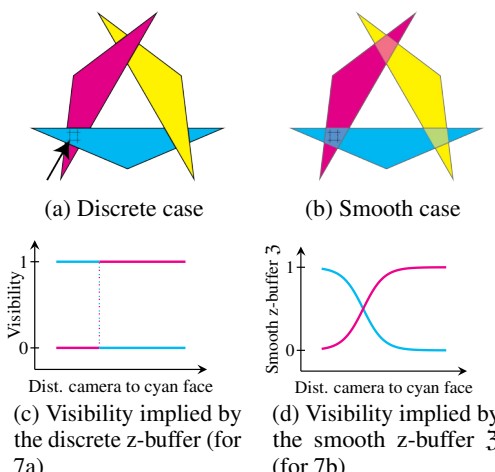

(a) Discrete case     (b) Smooth case

(c) Visibility implied by the discrete z-buffer (for 7a)

(d) Visibility implied by the smooth z-buffer $\mathfrak{z}$ (for 7b)

Figure 7: Visualization of the smooth depth buffer and occlusion: 7a shows three triangles rendered in a standard way, in 7b the same triangles are rendered smoothly. While in the discrete case a small change in depth can result in a sudden change of color (7c), our smooth depth-oriented rendering (7d) avoids that and therefore is differentiable everywhere.

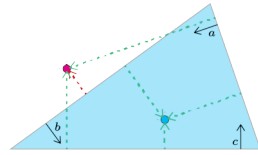 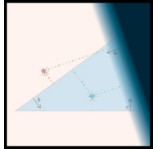 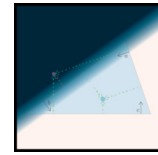 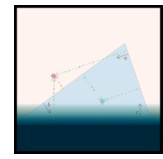 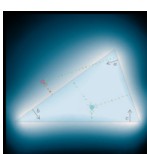

(a) check whether point is in triangle    (b) sigmoid wrt. edge $a$    (c) sigmoid wrt. edge $b$    (d) sigmoid wrt. edge $c$    (e) product of sigmoids

Figure 8: Visualization of the smooth rasterization. While the magenta point lies outside the triangle, the cyan point lies inside the triangle; this can be determined by measuring on which side of the edges a point lies. In subfigure 8b–8d it is smoothly determined which parts of the image lie inside and which parts lie outside the triangle with respect to the edges $a$–$c$. This is combined by multiplication (visibility tensor $V$) in subfigure 8e.

This smooth z-buffer leads to our $C^\infty$ Smooth Renderer, where the z-positions of triangles is differentiable with respect occlusions. In previous differentiable renderers, only the xy-coordinates were locally differentiable with respect to occlusion.

Let us assume to have three triangles (see Fig. 7a), where we want to examine the behavior of the bottom left pixel (marked with #) with respect to the z-position of the cyan face. During the process of optimizing the geometry, triangles might change their order with regard to the depth and abrupt color changes might appear. As shown in Figure 7c, the color value of the pixel (implied by the rasterization of the triangles) is constant except for one single point. At this point of intersection, the rasterization is discontinuous; at all other points, the derivative with respect to the z-position is 0. Employing the smooth rasterization and smooth z-buffer as in Figure 7d, the visibility of a pixel is never absolute, but rather a soft blend. Thus, it is differentiable, and optimizations can be solved with simple gradient descent.

Finally, we need to compute the color values of the triangles. For that, we use a lightning model composed of Blinn-Phong, diffuse, and ambient shading. We restrict the color to grayscale since we do not reconstruct the color in the RAN. Since the function of color is already differentiable we can directly use it.

Figure 9 shows a comparison between our smooth renderings and a Blender rendering of the Stanford bunny.

## A.1 SMOOTH Z-BUFFER $\mathfrak{Z}$

Our rasterization step is similar to the z-buffer algorithm, but instead of a displaying the single closet triangle and its z-distance in each pixel, we display a blend of triangles that project to the pixel.

We define the Smooth z-buffer $\mathfrak{Z}$ for pixel $(p_x, p_y)$, triangle $T$, and opacity $o$ as follows:

$$\mathfrak{Z}(p_x, p_y, T) := \mathfrak{w}\text{SoftMin}(o \cdot \text{z-dist}(\text{camera}, T), V(p_x, p_y, \cdot))$$

We define the weighted SoftMin (analogue to SoftMin/SoftMax) as: $\mathfrak{w}\text{SoftMin}(\mathbf{x}, w) := \mathfrak{w}\text{SoftMax}(-\mathbf{x}, w)$ where the weighted SoftMax is defined as:

$$\mathfrak{w}\text{SoftMax}_i(\mathbf{x}, w) := \frac{\exp(\mathbf{x}_i) \cdot w_i}{\sum_{i=0}^{\|w\|-1} \exp(\mathbf{x}_i) \cdot w_i} = \text{SoftMax}_i(\mathbf{x}_i + \log w_i)$$

Thus, for a pixel, the closest triangle is represented with high visibility, while triangles further away have weaker visibility. The visibility tensor $V$, as shall be defined in Section A.2, contains the extent to which a given triangle covers a given pixel. We use it as a weight for the $\mathfrak{w}\text{SoftMin}$, which allows considering only the relevant triangles in the SoftMin operator.

The opacity $o$ is a hyper-parameter setting accelerating the strength of the SoftMin. See Figure 9 to see how it affects the results.

Similar to the painter's algorithm (de Berg (1993)) we do not explicitly handle special cases of cyclical overlapping polygons that can cause depth ordering errors. Our smooth renderer is not

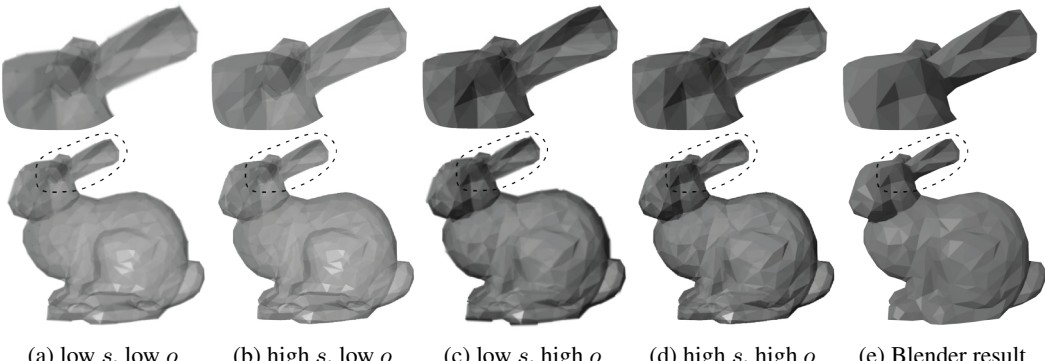

(a) low $s$, low $o$     (b) high $s$, low $o$     (c) low $s$, high $o$     (d) high $s$, high $o$     (e) Blender result

Figure 9: Stanford bunny rendered by the smooth renderer using different edge smoothnesses ($s$) and opacities ($o$): In 9a and 9b, the low opacity $o$ causes, e.g., one of the ears to be still visible through the head of the bunny (for usage of $o$ see section A.1). In 9a and 9c, the low edge steepness causes smoother edges (for usage of $s$ see section A.2). On the right: The Stanford bunny rendered by Blender (e).

sensitive to these cases. When polygons have a similar distance to the camera, their opacity will also be very similar and thus not only the front polygon but also the one behind is visible.

## A.2   VISIBILITY TENSOR $V$

In the general rendering approach, the discrete choice, whether a triangle covers a pixel is just a trivial check. In the smooth approach, as shown in Figure 8, we determine the pixels that correspond to a triangle by checking for each pixel whether the directed distances from the pixel to each edge are all positive. This yields the visibility tensor $V$ for triangle $T = (e_1, e_2, e_3)$ with $e_i = (v_1, v_2)$ and $v_i = (v_{i,x}, v_{i,y})$, and steepness $s$ as follows:

$$V(p_x, p_y, T) := \prod_{e=(v_1,v_2)\in T} \sigma \left( \begin{vmatrix} v_{x,2} - v_{x,1} & v_{x,1} - p_x \\ v_{y,2} - v_{y,1} & v_{y,1} - p_y \end{vmatrix} \cdot \frac{s}{m} \right)$$

$$\text{with } m = \text{SoftMin}_{e \in T}(\|e\|)$$

The sign of the directed distances to the three edges indicates on which side of the edge a pixel is. By applying a sigmoid function ($\sigma$) on that directed distances, we get a value close to 1 if the pixel lies inside and a value close to 0 if the pixel lies outside the triangle with respect to a given edge. By taking the product of the values for all three edges, the result ($\in [0, 1]$) smoothly indicates whether a pixel lies in or outside a triangle. Since this draws the triangles only from one direction, we add the same term with the negative directed distances to make the visibility tensor triangle orientation invariant:

$$V_{\text{orient.inv.}}(p_x, p_y, T) = \sum_{a \in \{-1,1\}} \prod_{e=(v_1,v_2)\in T} \sigma \left( a \begin{vmatrix} v_{x,2} - v_{x,1} & v_{x,1} - p_x \\ v_{y,2} - v_{y,1} & v_{y,1} - p_y \end{vmatrix} \frac{s}{m} \right)$$

For Figure 8, the visibility tensor $V$ looks as follows:

$$V(p_x/p_x, p_y/p_y, T) \overset{\text{example}}{=} \left( \rule{0pt}{0pt} \right) \cdot \left( \rule{0pt}{0pt} \right) \cdot \left( \rule{0pt}{0pt} \right)$$

The steepness $s$ is a hyper-parameter setting the steepness of the sigmoid function. See Figure 9 to see how it affects the results.

# B  3D GEOMETRY RECONSTRUCTIVE ADVERSARIAL NETWORK

Using our RAN framework, we train a reconstructor (pix2vex) without 3D supervision, i.e., without the need for the actual 3D models corresponding to the input images during training. The architecture of the reconstructor (pix2vex) is a pix2pix network followed by two fully connected layers predicting the coordinates of the points of a 3D mesh. The pix2vex shall be described in greater detail in Appendix D.2. The aforementioned causality dilemma is the following: To train the reconstruction, we need to compare the smoothly rendered images of the predicted shape to the input, which requires style transfer. On the other hand, to train the style translation, we need to know what a properly smoothly rendered image (corresponding to the input image) looks like. As discussed before, we solve the two components coevolutionarily by iteratively applying various influences towards a common solution.

The key idea is to train an adversarial discriminator $D$ to discriminate between the different ways to obtain pairs of images from $A$ (identity, p2v–$SR$–b2a, a2b–b2a) and $B$ (p2v–$SR$, a2b). This allows the three components p2v, a2b and b2a to be trained to fool $D$. In designing such a strategy, we exploited the following insights:

- Since pix2pix networks are lazy and their capabilities are restricted, the discriminator can be implicitly trained in a way that the content between pairs of images ($A$ and $B$) will be similar. The rationale behind this is the following: for the pix2pix to hold the cycle-consistency of p2v–$SR$–b2a, it is much easier for the image translator to only do a style-transfer from a content-wise similar image than to reconstruct the input from a smooth rendering of a different object.

- To let the discriminator know what a general smoothly rendered image should look alike, we train it by rendering randomly guessed 3D models. After doing so, the discriminator can be used to train a2b to output images from $B$.

Training a conventional GAN is a relatively straightforward task, since only a single binary decision (real vs. fake) has to be taken. Training the RAN, as shown in Fig. 4, is much more convoluted, since instead of only a single binary decision two decisions have to be made: one between three choices ($A$-input real vs. fake generated by p2v–$SR$–b2a vs. a2b–b2a) and one between two choices ($B$-input only fake generated by p2v–$SR$ vs. a2b). These paths represent all possibilities to obtain an image of $A$ respectively $B$—i.e., the discriminator has to differentiate between all possible ways to generate its input and thus fooling the discriminator leads to a common solution for all these paths.

Since it is easier to train a binary discriminator, our solution is to break the RAN into five sub-RANs, which all have to take only a single binary decision (real vs. fake), as depicted in Figure 11. These sub-RANs are alternately trained like conventional GANs by training their discriminator to differentiate between the "real" and "fake" input and training the other modules to fool the

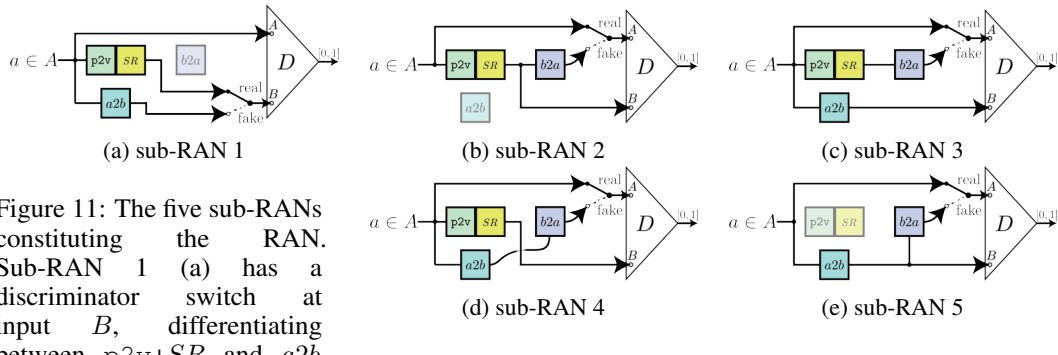

Figure 11: The five sub-RANs constituting the RAN. Sub-RAN 1 (a) has a discriminator switch at input $B$, differentiating between p2v+$SR$ and a2b. The other four routes differentiate between two images from space $A$, while the "real" input for the discriminator is always the input image. The "fake" input is the result of a round trip that either involves p2v+$SR$ and b2a (b, c) or a2b and b2a (d, e). This round trip is required for a cycle-consistency loss that diminishes mode collapse. Meanwhile, the $B$ input is the result of either p2v+$SR$ (b, d) or a2b (c, e). By that, p2v and a2b can be trained mutually.

Input                                    Prediction                              Alternative View

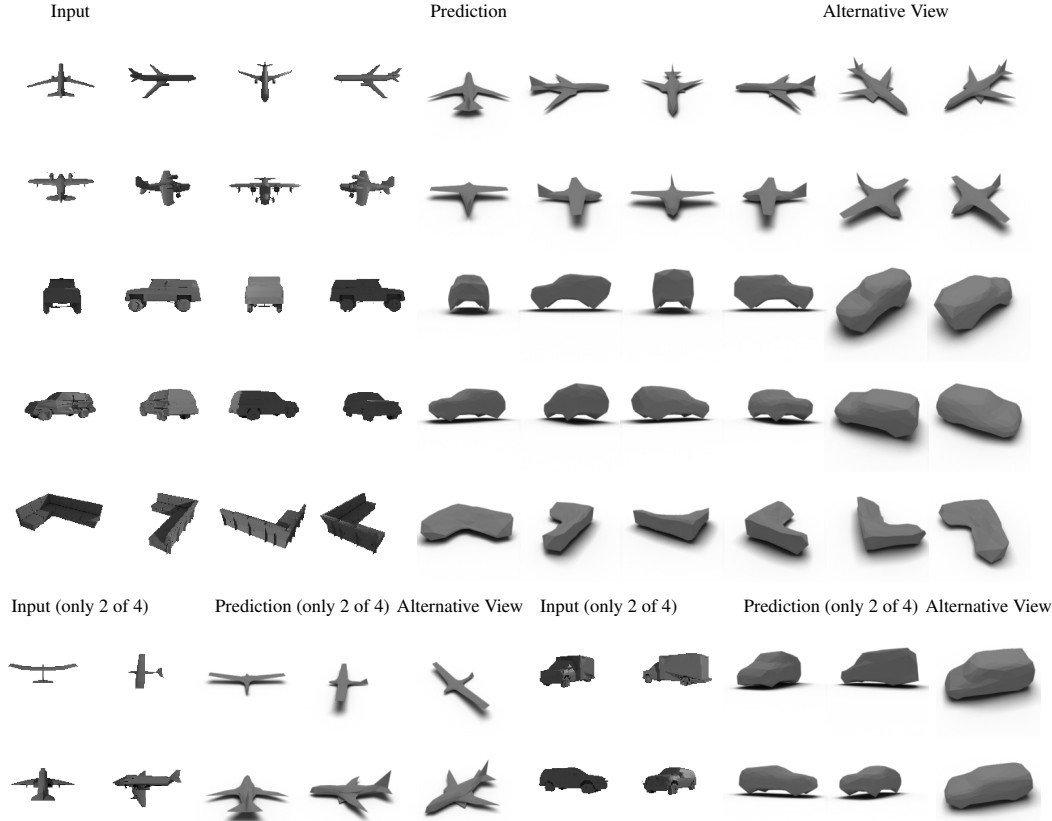

Input (only 2 of 4)      Prediction (only 2 of 4)  Alternative View  Input (only 2 of 4)      Prediction (only 2 of 4)  Alternative View

Figure 12: Four-view reconstruction of the ShapeNet (Chang et al. (2015)) classes *airplane*, *car*, and *sofa*. Left: input. Middle: prediction from the same angles. Right: predictions from alternative viewpoints.

discriminator. If there are two modules to be trained at once, the training is split into two steps: the module next to the discriminator is trained first and the one after the input is trained second (e.g., $b2a$, which is close to the discriminator, is trained first, and $a2b$ is trained second). This helps to avoid mode collapse. Since the relevance of these sub-RANs differs, their influence is weighted. For example, training a path with the `pix2vex` module (sub-RAN (a) in Figure 11) carries more weight than training the cycle of the two image translators (sub-RAN (e)). For the discriminator $D$, we use the binary cross-entropy loss. For training, an $L^1$ loss between any two images of the same image space is applied as defined in Section 3.3.

This constitutes the RAN as an unsupervised way to find an appropriate internal representation which in turn requires the `pix2pix` networks to perform a minimum of content-wise changes.

## C  RESULTS AND CONCLUSION

We evaluate our reconstruction results on synthetic as well as camera-captured images. While using synthetic images allows highly controlled experiments, the training and evaluation based on camera-captured images demonstrates that our approach can be applied to real-world scenarios.

For creating synthetic images, we used the ShapeNet dataset (Chang et al. (2015)) of categorized 3D meshes that has also been used for many other 3D reconstruction tasks. We rendered the 3D meshes using Blender with a resolution of typically $128 \times 128$ pixels and from multiple directions by using lighting hyperparameters different from the lighting hyperparameters that we used in the $SR$ of the RAN. This avoids unintended implicit supervision of the process.

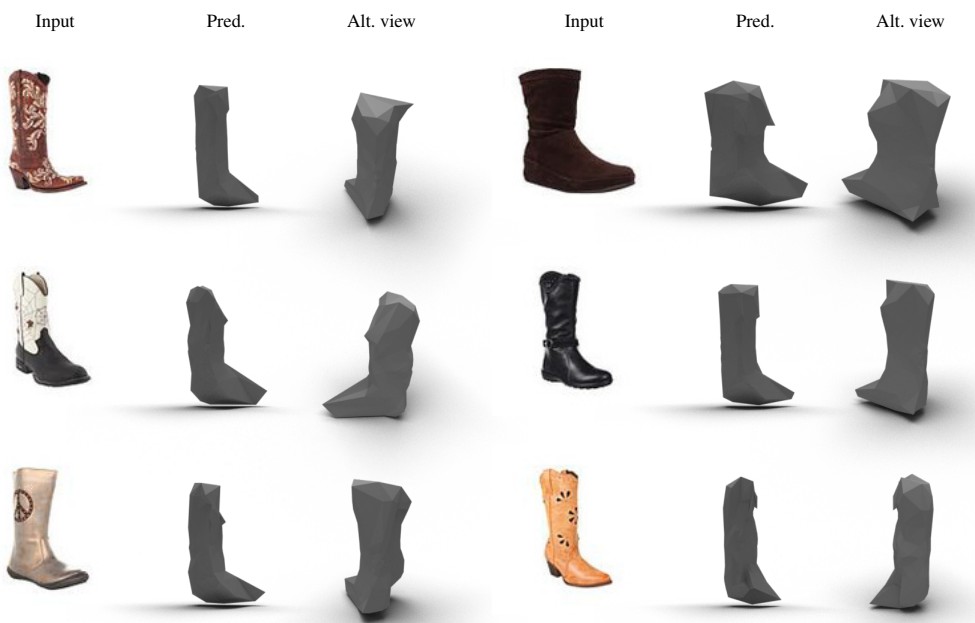

Figure 13: Single-view reconstruction results from the UT Zappos50K dataset (camera-captured images).

In the general case we used sets of images from four azimuths ($\Delta_{\mathrm{azimuth}} = 90°$) for our training. The results for this setting are shown in Fig. 12. In this case, the viewpoints are from altitudes which are not contained in the training data, e.g., the training images of cars were always from an altitude of $0°$—thus, a perfect reconstruction of a diagonal view of the models is harder. Since our smooth renderer does not consider shadows, reconstructing the sofa in line 5 is especially hard.

In addition, we conducted studies on modified settings as presented and described in Fig. 14. In (a–c), we performed four-view trainings with the following modifications to the training data: In (a), we randomized the azimuth of the images with a standard deviation of $5°$. In subfigure (b), we randomly assigned the position of the light source for each set of images. In subfigures (c1) and (c2), we trained on the *car* and *airplane* classes simultaneously.

In (d) and (e), we randomized, but supervised, the difference between azimuths for the four images. I.e., if multiple images have the same azimuth, the input data is effectively three or fewer images. In (d1) and (d2), we predicted the images from only two input images; in (e1–e4), we show single-view reconstructions. Since these reconstructions are trained on a single resp. dual view only, the quality of entirely unseen parts of the reconstruction is lower.

Training was performed on GTX Titan Xp GPUs on the basis of Float32. In our experiments, we used a uniform sphere with 162 vertices and 320 faces as base model. The networks for the presented results have been trained for between one and three weeks on a single GTX Titan Xp.

When processing low resolution images ($64 \times 64$ pixels) combined with a high mesh resolution (642 vertices, 1280 faces) faces have sub-pixel size. Thus, it occurs that single vertices dissociate themselves from the mesh since they are not visible any more. The problem of dissociating vertices is even worse if, instead of considering the directed distance to all edges of the faces only the distance to the faces is considered in the smooth visibility test.

For training on camera-captured images, we used single-view images of shoes (Yu & Grauman (2014)). Since these images are all typically taken from more or less the same direction, we use mirrored versions of the images and pretend this would be the view from the other side. We employed this small trick since many objects such as shoes are commonly roughly symmetric. Moreover, the back of the shoe could not be reconstructed without even having any training sample from the back side. Since this problem is highly ill-posed, our results could still be improved —nevertheless, they

are the first of their kind. Figure 13 presents the results of this single-view 3D mesh reconstruction which was trained on camera-captured images from a single direction.

In the proceeding experiments, we have demonstrated a robust way to reconstruct 3D geometry from only 2D data, eliminating the need for ground truth 3D models, or prior knowledge regarding materials and lighting conditions. In addition, we have demonstrated how a globally differentiable renderer is crucial to the learning process—even if designing one induces differences in the appearance of the produced renderings. We alleviate this difference through the use of image domain translation. The success of the reconstruction is driven by a restriction of the information flow and by the laziness of `pix2pix` networks, which easily perform image-style exchanges but struggle in changing the content of an image—a property that we exploit. Thus, our approach is not informed by data but instead by an understanding of the real world.

The presented experiments indicate that our RAN architecture is suitable to more than just its current application of 3D reconstruction, but instead to a variety of inverse problems.

In addition, given the right training dataset, we believe the performance of the RAN for camera-captured single-view image 3D reconstructions could be significantly improved.

## D    Implementation Details

### D.1    Regularization losses

The regularization losses ($\mathcal{L}_{\mathrm{reg}}$) on the reconstructed meshes with descending relevance are:

**The angle of normals**  of adjacent faces should be as similar as possible (loss uses the $L^2$ norm).

**The lengths of edges**  should be as similar as possible (loss uses the $L^1$ norm).

**The distance to the mean vertex**  of adjacent vertices should be as small as possible to imply a regular mesh and also reduce the curvature of the mesh (loss uses the $L^1$ norm).

### D.2    Network Architectures

Here, we describe the topologies of the components $p2v$, $a2b$, $b2a$, and $D$. In our experiments we typically used an image resolution of $n \times n = 128 \times 128$ and a number of vertices $v = 162$—we will base the following details on that assumption.

Let `Ck` denote a Convolution–LeakyReLU layer with k filters of size $4 \times 4$ and a stride of 2. Let the negative slope of the LeakyReLU be $0.2$. From the fifth convolutional layer on, we apply a $50\%$ dropout.

The **`pix2pix`** network is a symmetric residual network with the following blocks defining the first half: `C64-C128-C256-C512-C512-C512-C512`

**`pix2vex`** is based on the `pix2pix` network. It is followed by two fully connected layers ($n^2 \to \left\lfloor \frac{n^2 + 3 \cdot v}{2} \right\rfloor$ and $\left\lfloor \frac{n^2 + 3 \cdot v}{2} \right\rfloor \to 3 \cdot v$) and the sigmoid function.

***a2b*** and ***b2a*** are `pix2pix` networks with the first two or more residual layers. This is followed by the sigmoid function.

The **discriminator $D$** is defined as `C64-C128- C256-C512-C1` followed by the sigmoid function.

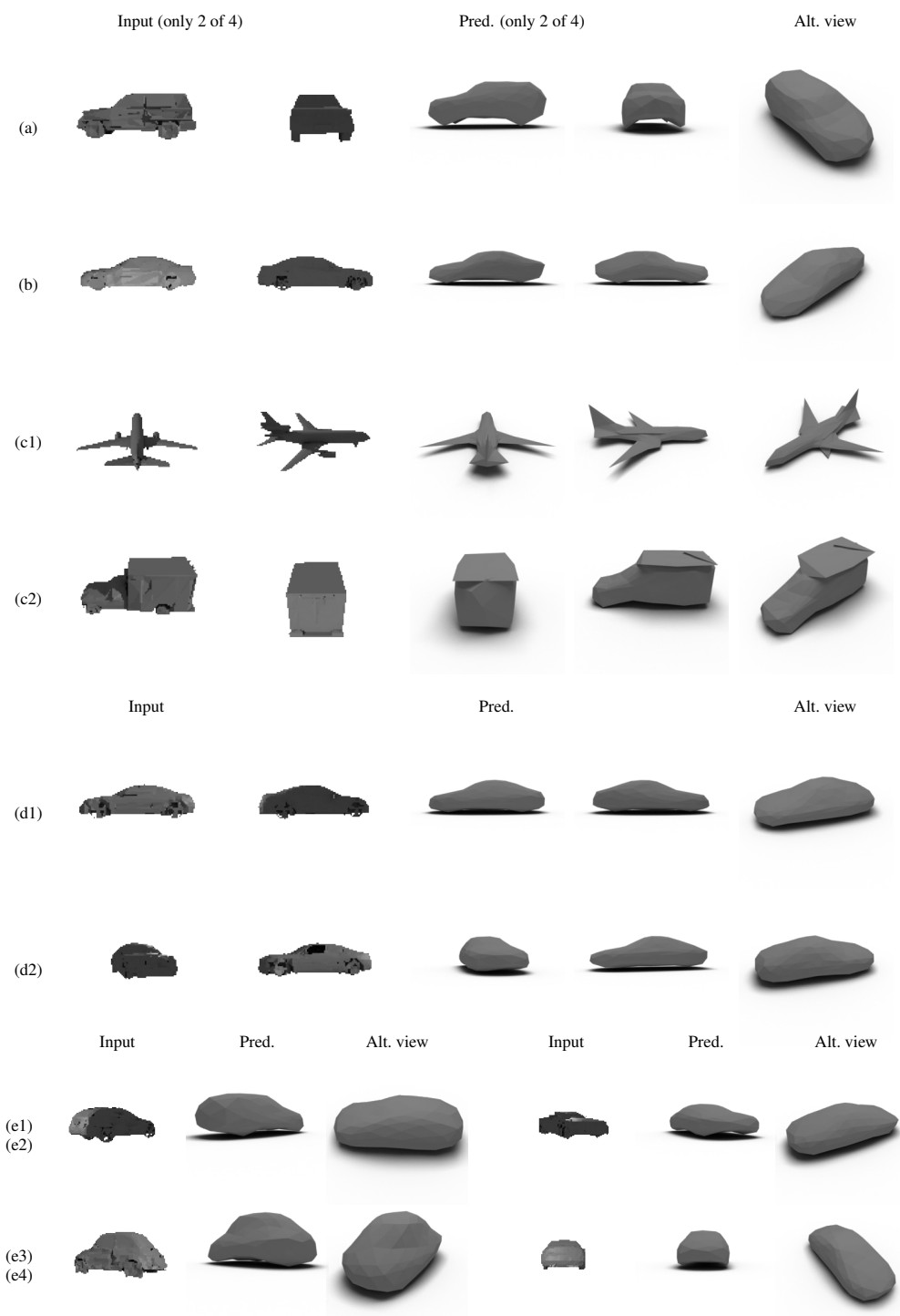

Figure 14: Experiments showing the robustness of our approach. (a–c): four-view training with the following modifications to the training data: (a) randomized azimuth of the images; (b) randomly assigned position of the light source; (c) simultaneous training of *car* and *airplane* classes; (d) predicting images from only two input images; (e) single-view reconstructions.

