# OpenReview forum: "AlgoNet: $C^\infty$ Smooth Algorithmic Neural Networks"
_ICLR.cc/2020/Conference — Reject_

### Official Review · AnonReviewer2 · 2019-10-23
**Official Blind Review #2**

**Rating:** 3

**Review:**

This paper presents a few different results surrounding smooth relaxations of classical algorithms. While the ideas (particularly the smooth rendering system) are interesting, the limited nature of the experiments and the lack of comparisons with baselines or prior work makes it difficult for me to support acceptance.

Some feedback:

- Re: "the output of C∞ smooth WHILE-programs differs from the discrete WHILE-programs by a small factor." Can you talk about the math here? Maybe mention bump functions (some of your relaxations have infinite support, and others have finite support, i.e. they use bump functions--this is likely to be an important distinction in practice).

- I'd want to see examples of softsort in action; especially examples that help me understand that the gradients are meaningful.

- The smooth renderer, and in particular the advantages it has over existing differentiable renderers, seem like the most important contributions of the paper (although they're relegated to the appendix). Can you expand on what you mean by "fully" vs "locally" differentiable? Can you provide empirical comparisons with other differentiable renderers?

- Can you optimize a scene with an unknown or variable number of triangles?

- I'm skeptical that all of the complexity of the RAN architecture and training setup is necessary. Is it possible to compare with other options, including baselines that leave out one or more of the components or training losses?

**Experience Assessment:**

I have published one or two papers in this area.

**Review Assessment: Checking Correctness Of Derivations And Theory:**

I assessed the sensibility of the derivations and theory.

**Review Assessment: Checking Correctness Of Experiments:**

I assessed the sensibility of the experiments.

**Review Assessment: Thoroughness In Paper Reading:**

I read the paper at least twice and used my best judgement in assessing the paper.

---

### Official Review · AnonReviewer3 · 2019-10-23
**Official Blind Review #3**

**Rating:** 1

**Review:**

The paper conceptualizes a neural network architecture that integrates smoothed versions of traditional algorithms into the network topology. The AlgoNet concept employs smoothed versions of algorithms implemented in the WHILE language, with options for different levels of differentiability. The paper outlines a forward version of AlgoNet based on traditional, skip, and residual connections, and a backward version for solving inverse problems in a reconstructive, autoencoder-like fashion. As smoothing introduces a form of domain shift, the paper the reconstructive adversarial network that that employs "domain translators" as well as a discriminator that are trained in an adversarial fashion. The paper concludes by describing versions of AlgoNet for various different algorithms.

The general idea of being able to better control the behavior of neural networks by better leveraging known structure (e.g., algorithms) is appealing. However, the paper does not go beyond conceptualizing how this might be done in a hand wavy manner. It is difficult to see what if anything can be learned from the paper, let alone what practical utility it has, which is important given that the paper claims to propose a new neural network architecture.

The paper would benefit from a discussion of empirical results in the main text, with baseline comparisons. With the exception of a single figure, the details are relegated to the appendices.

The related work discussion is surprisingly short, given the attention that has been paid to designing/optimizing different network topologies. The work that is discussed is very narrow in scope (see below).

The  paper devotes too much discussion to the challenges introduced by non-differentiable layers and the advantages of increasing degrees of differentiability (this is about half of the intro and most of the related work).

Despite the lack of experimental demonstrations, the paper is one page over the suggested 8 page limit.

**Experience Assessment:**

I have read many papers in this area.

**Review Assessment: Checking Correctness Of Derivations And Theory:**

I assessed the sensibility of the derivations and theory.

**Review Assessment: Checking Correctness Of Experiments:**

I assessed the sensibility of the experiments.

**Review Assessment: Thoroughness In Paper Reading:**

I read the paper at least twice and used my best judgement in assessing the paper.

---

### Official Review · AnonReviewer1 · 2019-11-11
**Official Blind Review #1**

**Rating:** 1

**Review:**

This paper describes "AlgoNets", which are differentiable implementations of classical algorithms. Several AlgoNets are described, including multiplication algorithm implemented in the WHILE programming language, smooth sorting, a smooth while loop, smooth finite differences and a softmedian.

The paper additionally presents RANs (similar to GANs but with an AlgoNet embedded) and Forward AlgoNets (where the the AlgoNet is embedded in a feedforward net).

The smooth implementations normally amount to replacing hard functions with soft equivalents, for example "if" conditions are replaced by logistic sigmoids.

The research direction in this paper is very interesting and could lead to important advancements, however a strong argument needs to be presented to the readers about why this way of making algorithms smooth is better than other published or obvious techniques.

The argument could be theoretical, proving for example faster convergence under certain assumptions, or it could be empirical, showing that the method achieves better results than other techniques on some benchmarks. I could not see however any such arguments in this paper.

**Experience Assessment:**

I have published in this field for several years.

**Review Assessment: Checking Correctness Of Derivations And Theory:**

I assessed the sensibility of the derivations and theory.

**Review Assessment: Checking Correctness Of Experiments:**

I assessed the sensibility of the experiments.

**Review Assessment: Thoroughness In Paper Reading:**

I read the paper at least twice and used my best judgement in assessing the paper.

---

### Decision · Program_Chairs · 2019-12-19

**Decision:**

Reject

**Comment:**

The paper does not provide theory or experiment to justify the various proposed relaxations. In its current form, it has very limited scope.